# Phenotypic and Molecular Characterization of *Candida albicans* Isolates from Mexican Women with Vulvovaginitis

**DOI:** 10.3390/jof11050354

**Published:** 2025-05-02

**Authors:** Hugo Díaz-Huerta, Eduardo García-Salazar, Xóchitl Ramírez-Magaña, Erick Martínez-Herrera, Rodolfo Pinto-Almazán, Paola Betancourt-Cisneros, Esperanza Duarte-Escalante, María del Rocío Reyes-Montes, Rigberto Hernández-Castro, María Guadalupe Frías-De-León

**Affiliations:** 1Programa de Maestría en Ciencias de la Salud, Escuela Superior de Medicina, Escuela Superior de Medicina, Instituto Politécnico Nacional, Plan de San Luis y Díaz Mirón s/n, Col. Casco de Santo Tomas, Alcaldía Miguel Hidalgo, Ciudad de México CP 11340, Mexico; hugo180893@gmail.com; 2Unidad de Calidad y Riesgo Biológico, Hospital Regional de Alta Especialidad de Ixtapaluca, Servicios de Salud del Instituto Mexicano de Seguro Social para el Bienestar (IMSS-BIENESTAR), Carretera Federal México-Puebla Km 34.5, Estado de México CP 56530, Mexico; 3Laboratorio de Micología Molecular, Unidad de Investigación Biomédica. Hospital Regional de Alta Especialidad de Ixtapaluca, Servicios de Salud del Instituto Mexicano de Seguro Social para el Bienestar (IMSS-BIENESTAR), Carretera Federal México-Puebla Km 34.5, Estado de México CP 56530, Mexico; eduardogs_01@hotmail.com (E.G.-S.); paola14_02@hotmail.com (P.B.-C.); 4Servicio de Ginecología y Obstetricia, Hospital Regional de Alta Especialidad de Ixtapaluca, Instituto Mexicano de Seguro Social para el Bienestar (IMSS-BIENESTAR), Carretera Federal México-Puebla Km 34.5, Estado de México CP 56530, Mexico; ramaxo@hotmail.com; 5Sección de Estudios de Posgrado e Investigación, Escuela Superior de Medicina, Instituto Politécnico Nacional, Plan de San Luis y Díaz Mirón s/n, Col. Casco de Santo Tomas, Alcaldía Miguel Hidalgo, Ciudad de México CP 11340, Mexico; erickmartinez_69@hotmail.com (E.M.-H.); rodolfopintoalmazan@gmail.com (R.P.-A.); 6Departamento de Microbiología y Parasitología, Facultad de Medicina, Universidad Nacional Autónoma de México, Avenida Universidad 3000, Ciudad Universitaria, Coyoacán, Ciudad de México CP 04510, Mexico; dupe@unam.mx (E.D.-E.); remoa@unam.mx (M.d.R.R.-M.); 7Department of Ecology of Pathogen Agents, Hospital Manuel Gea González, Calz. de Tlalpan 4800, Belisario Domínguez Secc 16, Alcaldía Tlalpan, Ciudad de México CP 14080, Mexico; rigo37@gmail.com

**Keywords:** vaginal infection, *Candida* spp., ABC genotyping, extracellular enzymatic activity

## Abstract

Vulvovaginal candidiasis (VVC) is an opportunistic mycosis that affects women of reproductive age. The most frequent etiological agent is *Candida albicans*. The development of VVC depends on factors related to the host and the fungus. Among the factors related to *Candida* spp. are virulence factors, but genotype may also be involved. The objective of this study was to evaluate the ABC genotypes and extracellular hydrolytic enzyme production in *C. albicans* isolates obtained from Mexican women with vulvovaginitis to determine if there is a correlation between these characteristics that allows the fungus to invade and cause damage to the host. Forty-three yeast isolates were obtained from vaginal exudates from women with symptoms of infection. The isolates were identified by germ tube tests and by Cand PCR. The ABC genotype of the isolates identified as *C. albicans* was determined through the isolates’ DNA amplification using the oligonucleotides CA-INT-R and CA-INT-L. The activity of esterase, phospholipase, proteinase, and hemolysin was evaluated in specific culture media. The correlation between extracellular enzyme production and genotype was analyzed using a two-way ANOVA and the Sidak comparison test. A total of 57.5% of the yeast isolates were identified as *C. albicans*. The genotypes identified were A (82.6%) and B (17.4%). The activity of esterase, phospholipase, proteinase, and hemolysin was very strong. No statistically significant difference was found between enzyme production and genotypes. In conclusion, genotype A predominates among *C. albicans* vaginal isolates. The production of extracellular hydrolytic enzymes was widely expressed in *C. albicans* vaginal isolates, but no correlation with genotype was found.

## 1. Introduction

Vulvovaginal candidiasis (VVC) is a localized mycosis caused by *Candida* spp., with *Candida albicans* being the most frequent species. VVC includes inflammation of the vagina and frequently involves the vulva [1]. The predominant symptom is vulvar pruritus, but abnormal vaginal discharge, vulvar burning, pain and irritation, dysuria, and/or dyspareunia may also occur [1]. It is worth mentioning that *Candida* is part of the normal microbiome, being identified in 10% of asymptomatic women [2], so a VVC diagnosis requires both the presence of *Candida* in the vagina and associated symptoms (irritation, burning, pruritus, dysuria, or inflammation) [3].

VVC represents a major health problem in the female population worldwide. It is estimated that about 75% of women of reproductive age suffer at least one episode of VVC throughout their lives, and of these, about 8% develop the infection recurrently, that is, four or more episodes of VVC in a year [4]. Several risk factors are involved in the development of VVC, which may be specific to the host or associated with their behavior, for example, alterations in the local defense mechanism, use of antibiotics, high glucose levels, sexual activity, high estrogen levels, genetic predisposition (presence of genetic polymorphisms), use of synthetic underwear, and use of tight clothing [3]. Likewise, some factors related to the ability of the fungus to change from its commensal state to pathogenic may also be involved. In this transition, some virulence factors allow the fungus to invade and cause damage to the host. Among the main virulence factors in *C. albicans* are the ability to undergo filamentation, which promotes increased adhesion and tissue invasion; biofilm formation, which leads to resistance to antifungals and host defenses, causing persistent infections; and extracellular hydrolytic enzyme production, such as hemolysin, phospholipase, esterase, and aspartyl protease, which cause damage to host tissues, facilitating penetration and invasion [5,6]. It has even been speculated that some *Candida* genotypes may be associated with mycosis development [7].

Genotyping of *C. albicans* has been performed using a variety of methods, including non-DNA-based methods (multilocus enzyme electrophoresis) and DNA-based methods (electrophoretic karyotyping, randomly amplified polymorphic DNA, amplified fragment length polymorphism, restriction enzyme analysis with and without hybridization, rep-PCR, microsatellite length polymorphism, multilocus sequence analysis, and ABC typing) [8]. Of these methods, ABC genotyping has been successfully used in the typing of *C. albicans* isolates from different sources and for different purposes [9,10,11,12,13,14,15,16,17]. This method detects the presence or absence of an intron in the 25s rDNA by multiple PCR, and depending on the size of the amplicon generated, *C. albicans* is classified into four genotypes: genotype A (450 bp), genotype B (840 bp), genotype C (450 and 840 bp), and genotype E (1000 bp) [18]. ABC genotyping is a highly reproducible method with low cost, little methodological complexity, and high discriminating power.

Some typing studies have shown that, depending on the geographical region, certain *C. albicans* genotypes predominate in VVC, which may be related to a more severe symptomatology [19]. Therefore, it is critical to elucidate whether there is an association between genotype and virulence attributes, as this can be useful to design effective strategies for controlling and treating VVC in specific populations [14]. Therefore, the objective of this study was to evaluate the extracellular enzyme activity and the ABC genotype of *C. albicans* isolates obtained from Mexican women with vulvovaginitis and to determine if there is a correlation between these characteristics.

## 2. Materials and Methods

### 2.1. Study Design

An observational, analytical, cross-sectional, and prospective study was conducted in a tertiary-care hospital in Mexico.

### 2.2. Study Population

This study included adult women who received care at the Obstetrics and Gynecology Unit of the High Specialty Regional Hospital of Ixtapaluca (HRAEI) from October 2022 to December 2023 and consented to participate. The research protocol was approved by the HRAEI Research and Research Ethics Committees (NR-CEI-HRAEI-12-2022).

### 2.3. Inclusion Criteria

The inclusion criteria were adult women (≥18 years), not pregnant, with signs or symptoms of VVC: vaginal discharge with no odor, pruritus, burning, dysuria, edema, erythema, and/or dyspareunia.

The criteria for the diagnosis certainty of VVC was a positive culture for *Candida* sp. and the presence of one sign or symptom [20].

### 2.4. Exclusion Criteria

Women who were under antifungal treatment or had completed treatment in the previous month and those with immunosuppressive diseases were excluded from this study.

Women with non-*albicans Candida* isolates from vaginal exudate samples were also excluded.

### 2.5. Sampling

A sample of cervicovaginal exudate (posterior sac fundus and cervix) was taken with a sterile swab with a cotton tip and Stuart transport medium. The samples were taken by the treating gynecologist.

### 2.6. Yeast Isolation

The swab with the clinical sample was inoculated on plates with Sabouraud Dextrose Agar (SDA) (BD Bioxon, Mexico City, MX). Cultures were incubated at 37 °C for 24–72 h. All the yeast isolates obtained were subcultured in SDA (Bioxon) and incubated at 37 °C for 24–48 h in an aerobic atmosphere.

### 2.7. Phenotypic Identification of C. albicans

#### Germ Tube Test

A colony yeast was taken and resuspended in 500 μL of human serum and incubated at 37 °C for 2 h. Subsequently, 20 μL of serum was placed between a slide and a coverslip and observed at 40x under an optical microscope to determine the presence or absence of germ tubes (elongated daughter cells from the round mother cell without constriction at their origin). The criterion for germ tube positivity was the observation of at least five germ tubes in the entire wet mount. Negative results were confirmed by examining at least 10 fields to exclude the presence of germ tubes [21].

### 2.8. DNA Extraction

The lithium acetate method was used [22]. The yeasts were resuspended in 100 μL of a 200 mM lithium acetate (LiOAc) 1% SDS solution and incubated at 70 °C for 15 min. An amount of 300 μL of 96% ethanol was added and subjected to vortex agitation for 10 s, then centrifuged at 15,000 g for 3 min, and the supernatant was decanted. The sediment was washed with 70% ethanol, the supernatant was decanted, and the button was allowed to dry for 12 h. Finally, 100 μL of DNase-free water was added to dissolve DNA. The debris was removed by centrifugation at 15,000× *g* for 1 min, and 1 µL of the supernatant was used for PCR. The quality and quantity of the DNA obtained were analyzed by spectrophotometry (DS 11 Spectrophotometer, DeNovix Inc., Wilmington, USA) at 260 and 280 nm.

### 2.9. Molecular Identification of C. albicans

Vaginal yeast isolates were identified by the Cand PCR assay, described by García-Salazar et al. [23]. Reactions were conducted in a final volume of 25 μL containing 10 ng of DNA, 200 μM dNTP (Jena Bioscience, Jena, GE), 1.5 mM MgCl_2_, 100 pmol of each oligonucleotide (Cand-F and Cand-R) (Sigma-Aldrich, Saint-Louis, USA), 1 U *Taq* DNA polymerase (Jena Bioscience), and 1X PCR buffer. As a positive control, 10 ng of DNA from the reference strain *C. albicans* ATCC^®^ 18804™ was amplified, and sterile deionized water (Milli-Q) was included as a negative control. The amplification program consisted of 1 cycle of 3 min at 94 °C, 33 cycles of 30 s at 95 °C, 30 s at 55 °C, 1 min at 72 °C, and a final extension cycle of 5 min at 72 °C. At the end of the reaction, 5 μL of the amplification products were analyzed by electrophoresis in 1.5% agarose gel (Axygen BioScience, Madison, WI, USA) and stained with Midori Green (NIPPON Genetics EUROPE, Düren, Germany) in TBE buffer 0.5X (Tris-Base 45 mM, boric acid 45 mM, EDTA 1 mM, pH 8.3) at 70 V. The 100 bp DNA Ladder (Promega, Madison, WI, USA) was used as a molecular size marker. Images of the gels were captured using a gel documentation system (Cleaver Scientific Ltd., UK). The molecular identification of yeasts was determined according to amplicon size: *C. albicans* (850 bp), *Nakaseomyces glabratus* (formerly *C. glabrata*) (1000 bp), *C. tropicalis* (790 bp), *C. parapsilosis* (731 bp), *Pichia kudriavzevii* (formerly *C. krusei*) (800 bp), *Meyerozyma guilliermondii* (formerly *C. guilliermondii*) (1100 bp), *Clavispora lusitaniae* (formerly *C. lusitaniae*) (590 bp), and *C. dubliniensis* (810 bp). The isolates identified as *C. albicans* were characterized genotypically and phenotypically.

### 2.10. Genotypic Characterization

#### ABC Genotyping

*C. albicans* vaginal isolates were typed by the ABC method [13]. Amplification reactions were carried out in a final volume of 25 μL, containing 100 ng of genomic DNA, 0.2 mM of each dNTP (Jena Bioscience, Jena, Germany), 0.5 mM of each oligonucleotide (CA-INT-L 5′ ATAAGGGAAGTCGGCAAAATAGATCCGTAA-3′ and CA-INT-R 5′− CCTTGGCTGTGTTTCGCTAGATAGAT-3′) (Sigma-Aldrich), 1.5 mM of MgCl_2_, 1 U Taq polymerase (Jena Bioscience), and 1X PCR buffer. The amplification program consisted of a 3 min cycle at 95 °C, followed by 30 cycles of 30 s at 94 °C, 30 s at 60 °C, and 40 s at 72 °C, and a final extension cycle of 5 min at 72 °C. The amplified fragments were analyzed by electrophoresis in 1.5% agarose gel stained with Midori Green (NIPPON Genetics EUROPE) in a 0.5X TBE buffer. The 100 bp DNA Ladder (Promega) was used as a molecular size marker. Images of the gels were captured using a gel documentation system (Cleaver Scientific Ltd. Rugby, UK).

### 2.11. Phenotypic Characterization

#### Hydrolytic Enzyme Activity

*C. albicans* vaginal isolates were phenotyped by evaluating the production of extracellular hydrolytic enzymes (hemolysin, aspartyl protease, phospholipase, and esterase) [24]. The following media containing specific substrates were used: egg yolk for phospholipase activity and bovine serum albumin (Sigma-Aldrich) for proteinase activity. SDA supplemented with defibrinated sheep blood (dDBiolab, Barcelona, ES) and 3% glucose (BD Bioxon) was used to evaluate hemolysin activity. The esterase activity was evaluated in culture media with yeast extract (BD Bioxon) and peptone (BD Bioxon) supplemented with Tween 80 at 0.5% (Sigma-Aldrich). Lastly, phospholipase and hemolysin activity were evaluated in cultures incubated at 37 °C, while esterase and proteinase activity were incubated at 25 °C. After the incubation period, the colony diameter and clear zone were measured. All trials were performed in triplicate and on two different occasions using the *C. albicans* reference strain ATCC^®^ 18804™ as a positive control. The enzyme activity index or precipitation zone (Pz) was determined as described by Erum et al. [25] using the formulaPz = colony diameter (mm)/[colony diameter (mm) + halo diameter (mm)]

The Pz can take values ranging from 0 to 1. Values less than 0.69 are considered very strong activity, values between 0.70 and 0.79 are considered moderate activity, those between 0.80 and 0.89 are considered weak activity, those between 0.9 and 0.99 are considered very weak, and a value equal to 1 is considered inactive [25].

### 2.12. Statistical Analysis

Descriptive statistics were applied to the qualitative variables, using frequencies and percentages to analyze the obtained data. Regarding the quantitative variables, the Kolmogorov–Smirnov normality test was performed (n > 30), and the mean and standard deviation were used, as the data had a normal distribution. To determine whether there were differences between the enzymatic activity (hemolysin, esterase, phospholipase, and proteinase) in the isolates in general, the one-way ANOVA test with Tukey’s post hoc test was used. To determine if there was a relationship between the enzymatic activity (hemolysin, esterase, phospholipase, and proteinase) and the genotypes (A and B), the two-way ANOVA test was used, followed by the Sidak comparison test. The GraphPad Prism software version 9, CA, USA, was used for statistical analysis.

## 3. Results

During the study period, 243 women met the inclusion criteria and consented to participate. Of the 243 vaginal exudate samples obtained, yeasts were isolated in 43 (17.7%), bacterial growth was observed in 111 (45.7%), and no development of microorganisms was found in 89 (36.6%) (Figure 1). The combined growth of yeast and bacteria was not observed in any sample. The isolates were named numerically (01 to 243) based on the consecutive number of the clinical sample.

### 3.1. Identification of C. albicans

The 43 yeast isolates presented white, smooth, and creamy colonies, macromorphological features typical of the *Candida* genus.

#### Germ Tube Test

Of the 43 yeast-like isolates, germ tube development was observed in only 23 (53.5%) (Figure 2).

### 3.2. Molecular Identification

The amplicons of 850 and 1000 bp, generated by Cand PCR, showed that 23 isolates (011, 015, 023, 032, 045, 078, 097, 100, 107, 124, 135, 149, 151, 163, 169, 183, 192, 202, 216, 217, 218, 234, and 243) corresponded to *C. albicans* and 20 to *N. glabratus* (Figure 3).

### 3.3. Genotypic Characterization

#### ABC Genotyping

Of the 23 vaginal isolates of *C. albicans*, 19 (011, 015, 032, 045, 078, 097, 100, 107, 124, 135, 151, 169, 183, 192, 202, 216, 217, 234, and 243) amplified a 450 bp fragment, showing that they belong to genotype A, while 4 (023, 149, 163, and 218) isolates amplified an 840 bp fragment, corresponding to genotype B. No isolates with genotypes C or E were detected (Figure 4).

### 3.4. Phenotypic Characterization

#### Hydrolytic Enzyme Activity

The secretion of hydrolytic enzymes was variable among the 23 isolates identified as *C. albicans*. In the case of hemolysin, all 23 *C. albicans* isolates showed evidence of enzyme secretion in the culture medium. The Pz of hemolysin ranged from 0.25 to 0.38, i.e., the isolates had very strong hemolysin activity (Table 1).

Esterase secretion was observed in 17 (73.9%) isolates, while 6 isolates did not present an inhibition halo in the culture medium. The Pz of esterase was in the range of 0.29–0.43, which revealed very strong esterase enzyme activity in most of the vaginal isolates analyzed (Table 1).

Phospholipase secretion was detected in 22 isolates. One isolate did not show production of the enzyme in vitro. The Pz of phospholipase presented values of 0.29–0.47. These values indicated very strong phospholipase activity in the isolates (Table 1).

In the case of proteinase, 100% (23) of the *C. albicans* isolates showed proteinase secretion with Pz between 0.32 and 0.54, with very strong activity (Table 1).

### 3.5. Statistical Analysis

Among the *C. albicans* isolates obtained from patients with CVV, a statistically significant difference was observed in enzyme production, with the production of hemolysins being lower than that of the other enzymes (Figure 5a). However, when performing the two-way ANOVA with the Sidak post hoc test, no statistically significant differences were observed in terms of enzyme production between genotypes A and B (*p* > 0.5031), maintaining the statistical difference between the production of the different types of enzymes (*p* > 0.001). Furthermore, no statistically significant difference was observed in the interaction between genotypes A and B and the types of enzymes studied (*p* > 0.2535).

These results denote no statistically significant difference between enzyme production and genotypes (Figure 5b). Therefore, there is no correlation between the ABC genotype and the extracellular hydrolytic enzyme activity indexes (hemolysin, phospholipase, proteinase, and esterase), and the isolate’s genotype does not influence their production.

## 4. Discussion

VVC is the second most common vaginal infection after bacterial infection [26]. VVC is an infection that does not pose a risk to the lives of those who suffer from it; however, it constitutes a relevant public health problem due to the size of the affected population, the negative impact on women’s sexual and social lives, and the high cost it represents for health institutions, as it is a frequent reason for gynecological consultations [27,28]. It is estimated that by 2030, the population of women with vulvovaginal candidiasis will increase, causing an economic burden of up to USD 14.39 billion annually associated with the loss of productivity of affected women [4]. VVC in patients with diabetes mellitus is also a frequent complication that generates indirect costs [29]. In addition, the complications that can occur if VVC is not treated correctly make it a significant health problem. These complications include pelvic inflammatory disease, infertility, ectopic pregnancy, pelvic abscess, miscarriage, and menstrual disorders, among others [1].

*C. albicans* is the most frequent agent in the etiology of VVC, causing about 85–90% of cases [2]. This fungus is a common colonizer of human skin and mucous membranes. Therefore, to develop an infection like VVC, aspects like *Candida’s* ability to modulate the expression of virulence factors in response to microenvironmental changes and the host’s immune status are involved [30]. Among the virulence factors are filamentation and biofilm formation (hyphal forms coated by an extracellular polymeric substance), which provide them with a protective layer [6]. Combining these two factors allows *C. albicans* to adhere, colonize, resist antifungal treatments, and penetrate host tissues, contributing to its virulence and ability to cause persistent infections [31]. The extracellular enzyme secretion is an important virulence factor, as it facilitates tissue invasion and inactivation of the host immune system components [32]. The secretion of phospholipases helps yeasts to degrade glycerolphospholipids, a component of the host cell membrane, which leads to the destruction of cells and the usage of nutrients for their growth and infection development [33]. In addition, it is believed that the genotype of the fungus may also be involved in the pathogenesis of VVC [8,19]. Therefore, conducting studies to gain more knowledge about the epidemiology of VVC is necessary. In this sense, this study determined the ABC genotype of *C. albicans* vaginal isolates, as well as the production and activity of hemolysin, esterase, phospholipase, and proteinase. Likewise, the possible correlation between genotype and virulence factors was analyzed.

From a sample of 243 non-pregnant adult women with symptoms compatible with VVC, we found an incidence of 17.7% (43/243), which is within the range of incidence (12.1% to 57.3%) reported in women of reproductive age in different countries [34]. As etiological agents of VVC, only two species were found, with *C. albicans* being the most frequent at 53.5% (23/43), followed by *N. glabratus* at 46.5% (20/43). This species distribution is consistent with that reported in other geographical regions, where the most common non-albicans species is *N. glabratus* [34,35]. However, some studies, such as that of Amanloo et al. [11], reported *N. glabratus* (56.1%) as the primary agent of VVC, followed by *C. albicans* (39%). The detection of non-albicans species as causal agents of VVC shows the need to identify the fungus at the species level when diagnosing this mycosis. The latter is relevant because non-*albicans* species tend to have a different antifungal susceptibility profile than *C. albicans*, which impacts treatment and, possibly, the severity of the clinical picture [35]. A straightforward way to discriminate *C. albicans* from *C.* non-*albicans* is by observing germ tube development [36]. However, this technique cannot be used independently. It must be combined with another technique, as it has been reported that other species, such as *C. tropicalis*, can develop such a structure [37]. The germ tube test identified 23 isolates as *C. albicans* in this study. Molecular assay (Cand PCR) confirmed this result, which showed the 850 bp amplicon corresponding to this species [23].

In the genotyping of *C. albicans* vaginal isolates, we found a predominance of genotype A (82.6%) over genotype B (17.4%), while genotypes C and D were not detected. This genotype distribution is consistent with the findings of Amanloo et al. [12], who typed *C. albicans* vaginal isolates obtained from health centers in the city of Zanjan, Iran. Their results showed that genotype A was the most common (43.7%), followed by B (31.3%) and C (25%). Genotype E was not detected. In the same way, Kumar and Tejashree [38] genotyped 282 *C. albicans* isolates, finding that genotype A was predominant (87.6%) over genotypes B (9.9%) and C (0.2%). Genotype A has also been the most frequent genotype in both *C. albicans* isolates from patients with HIV, patients with invasive candidiasis, and the mycoflora of healthy individuals [39,40,41]. Even the typing of *C. albicans* isolates obtained from animals presents the same genotype distribution [18]. In contrast, another study conducted at the Ahvaz Jundishapur University of Medical Sciences in Iran reported that of 103 *C. albicans* isolates, the predominant genotype was C (83.5%), followed by B (8.7%) and A (3.9%). These results indicate that the genotype distribution may depend on the isolates’ geographical origin and the population type [42].

On the other hand, it has been suggested that the extracellular enzymatic activity in *C. albicans* is a characteristic of the fungus that may depend on the source of the isolate [43]. In this study, the totality of *C. albicans* isolates from vaginal samples had strong hemolysin activity, regardless of the genotype. This may be related to the high virulence of the studied isolates, since it has been reported that only *C. albicans* hyphae use hemoglobin as a source of iron, and in turn, filamentation is a crucial virulence factor in CVV. [44]. The high hemolysin activity found is consistent with the findings of Fatahinia et al. [45], who analyzed the activity of enzymes secreted by *Candida* spp. vaginal isolates and found that all isolates secreted hemolysin. However, this contradicts that reported by Gharaghani et al. [13], who determined that only 40.8% of 103 *C. albicans* isolates evaluated secreted hemolysin and that the variability in the enzyme activity correlated with the different genotypes.

In the case of esterase, we observed that 73.9% of the isolates secreted this enzyme, showing a strong activity but without correlation with genotype A or B. Esterases hydrolyze lipid compounds of host cell membranes, thereby causing cell dysfunction and disruption, and they likely facilitate *Candida* adherence and tissue penetration [46]. Therefore, the isolates studied present a high virulence potential that is not related to the genotype; however, this must be corroborated in subsequent studies involving a larger number of samples. In contrast to our findings, Gharaghani et al. [13] reported that the majority (95.2%) of the *C. albicans* isolates they analyzed could secrete esterase, with an activity index in the range of strong to moderate, depending on the genotype.

For phospholipase, 95.6% of the isolates included in this study showed secretion of this enzyme with strong activity but without a correlation with the genotype. In contrast, another study reported that 84.8% of *C. albicans* vaginal isolates secreted phospholipase with strong activity associated with the genotype [13].

Finally, strong proteinase activity was observed in 100% of the vaginal isolates studied, but no correlation with genotypes was detected. Proteases participate in the initial steps of colonization by damaging host tissues and digestion of the host proteins, such as collagen, keratin, and mucin, to generate nutrients for *Candida*, which helps the fungus invade the vaginal epithelium more deeply [47]. Therefore, the proteinase activity observed in the analyzed isolates suggests a high virulence. Since genes that regulate proteinase production (*SAP*s) have been strongly expressed in women with symptomatic vaginitis, but not in those with asymptomatic colonization [48], there may be a certain relationship between genotypes A and B, which are the ones found in this study, and virulence. However, more studies are required to corroborate this. The strong proteinase activity in the isolates studied is consistent with what was reported by Gharaghani et al. [13], who observed in vitro secretion of this enzyme in 97.1% of *C. albicans* vaginal isolates without showing a significant difference between the enzyme production in each genotype.

The lack of correlation between the *C. albicans* genotype and other characteristics, such as host type, anatomical site of isolation, and susceptibility to antifungals, has been reported by Davland et al. [18] when typing animal-derived *C. albicans* isolates. However, in the present study, the lack of correlation found between enzymatic secretions and the ABC genotypes of *C. albicans* isolates may be due to the small sample size, so further studies are recommended to investigate the molecular relationship between the genotype and virulence of *C. albicans*. Nonetheless, the results of this study corroborate the variability in the enzyme secretion of *C. albicans*, depending on the body site where the yeasts were isolated. Yeasts of vaginal origin showed high production and strong activity of extracellular enzymes, in contrast to yeasts from esophageal samples, which have low production and activity of hemolysin (68.4%), phospholipase (47.4%), and proteinase (36.8%) [11].

### Limitations

One of this study’s main limitations is the reduced number of studied isolates, which may interfere with the correlation analysis between genotype and extracellular hydrolytic enzyme secretions. Another limitation is that the isolates were collected from a single hospital in Mexico, so it is not possible to generalize conclusions regarding the predominance of genotype A in the geographical region nor the type of sample from which the yeasts were isolated.

ABC genotyping has proven to be a good typing method; however, we acknowledge that there are other molecular biology techniques, such as high-throughput sequencing, that offer a more accurate typing of *Candida* and allow the rapid and efficient identification and characterization of genes associated with fungal virulence [49,50].

## 5. Conclusions

In the *C. albicans* isolates obtained from Mexican women with vulvovaginitis, genotype A predominates, and the production of hemolysin, esterase, phospholipase, and proteinase is widely expressed; however, no correlation was found between the enzymatic activity and the ABC genotype of the isolates.

## Figures and Tables

**Figure 1 jof-11-00354-f001:**
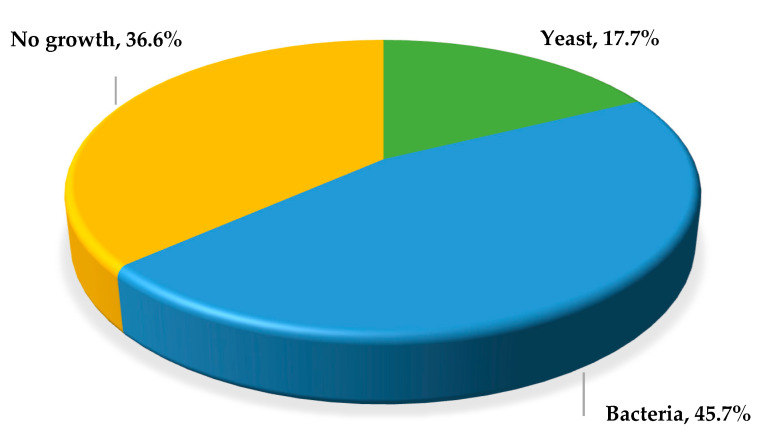
Frequency and type of microorganisms isolated from vaginal exudate samples from women with vulvovaginitis.

**Figure 2 jof-11-00354-f002:**
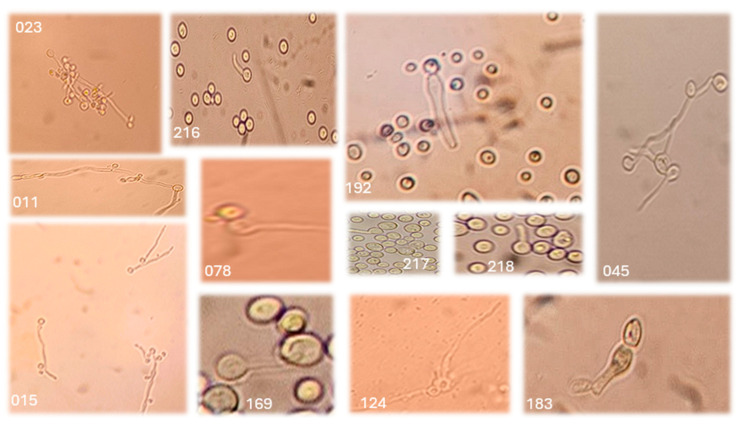
Germ tube development in isolates (011, 015, 023, 045, 078, 124, 169, 183, 192, 216, 217, and 218) of *Candida albicans* obtained from vaginal samples.

**Figure 3 jof-11-00354-f003:**
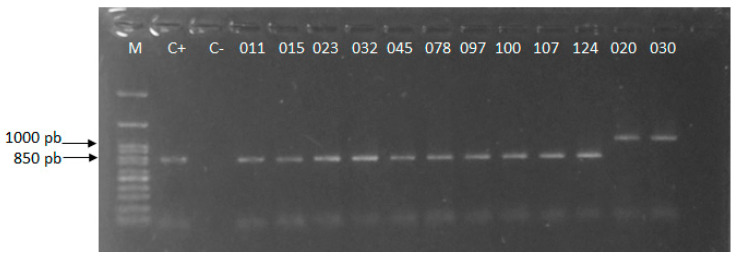
Molecular identification of yeasts isolated from vaginal samples. The figure shows the amplicons generated by Cand PCR, analyzed by electrophoresis in 1.5% agarose gel in TBE 0.5X buffer, and stained with Midori Green. M: 100 bp size molecular marker. C-: negative control, C+: positive control (DNA of *Candida albicans* ATCC^®^ 18804™).

**Figure 4 jof-11-00354-f004:**
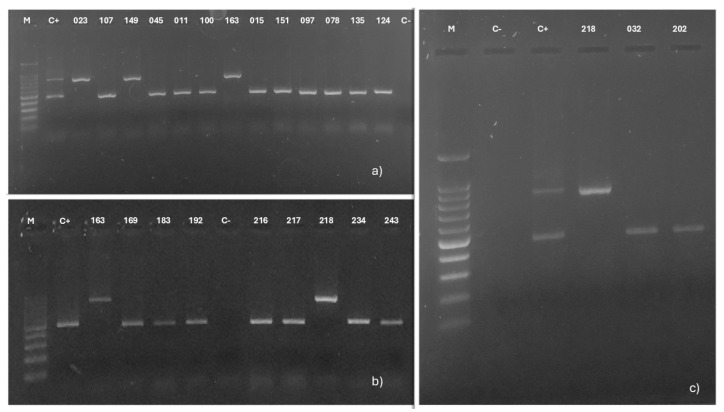
ABC genotyping of 23 *Candida albicans* isolates obtained from vaginal samples. (**a**–**c**) show the amplicons generated with the CA-INT-R and CA-INT-L oligonucleotides, analyzed by electrophoresis using 1.5% agarose gel in a 0.5X TBE buffer, and stained with Midori Green. M: 100 bp size molecular marker. C-: negative control, C+: in (**a**,**c**), DNA from an isolate with genotype C was used, and in (**b**), DNA from an isolate with genotype A was used.

**Figure 5 jof-11-00354-f005:**
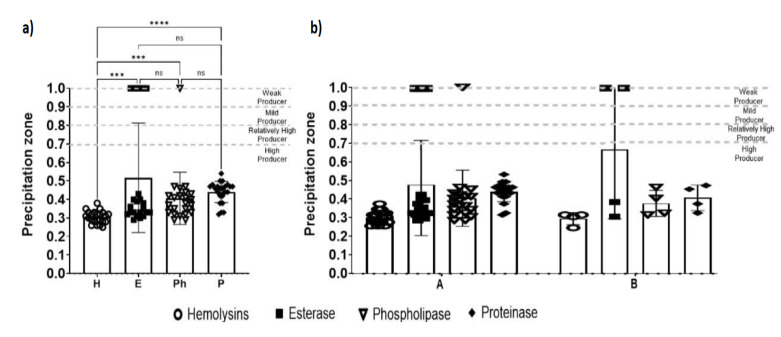
(**a**) Analysis of enzymatic activity in the isolates and (**b**) analysis of enzymatic activity with respect to the ABC genotype of *Candida albicans* isolates obtained from patients with CVV. *** *p* < 0.001; **** *p* < 0.0001.

**Table 1 jof-11-00354-t001:** Genotype and hydrolytic enzyme activity in *C. albicans* vaginal isolates.

Isolate	Genotype	Enzyme Activity Index (Pz)			
		Hemolysin	Esterase	Phospholipase	Proteinase
011	A	0.28 ^b^	0.31 ^b^	0.47 ^b^	0.42 ^b^
015	A	0.38 ^b^	0.30 ^b^	0.43 ^b^	0.38 ^b^
032	A	0.29 ^b^	0.29 ^b^	0.29 ^b^	0.47 ^b^
045	A	0.29 ^b^	0.36 ^b^	0.44 ^b^	0.46 ^b^
078	A	0.29 ^b^	1.00 ^a^	0.46 ^b^	0.44 ^b^
097	A	0.31 ^b^	0.32 ^b^	0.40 ^b^	0.44 ^b^
100	A	0.33 ^b^	0.35 ^b^	0.42 ^b^	0.47 ^b^
107	A	0.28 ^b^	0.38 ^b^	0.35 ^b^	0.46 ^b^
124	A	0.32 ^b^	0.33 ^b^	0.33 ^b^	0.33 ^b^
135	A	0.35 ^b^	0.33 ^b^	0.41 ^b^	0.32 ^b^
151	A	0.30 ^b^	1.00 ^a^	0.32 ^b^	0.42 ^b^
169	A	0.32 ^b^	0.40 ^b^	1.00 ^a^	0.48 ^b^
183	A	0.33 ^b^	0.43 ^b^	0.31 ^b^	0.50 ^b^
192	A	0.35 ^b^	0.33 ^b^	0.29 ^b^	0.47 ^b^
202	A	0.26 ^b^	1.00 ^a^	0.35 ^b^	0.47 ^b^
216	A	0.26 ^b^	0.33 ^b^	0.37 ^b^	0.44 ^b^
217	A	0.33 ^b^	0.33 ^b^	0.37 ^b^	0.47 ^b^
234	A	0.28 ^b^	1.00 ^a^	0.42 ^b^	0.50 ^b^
243	A	0.26 ^b^	0.40 ^b^	0.38 ^b^	0.54 ^b^
023	B	0.31 ^b^	0.31 ^b^	0.33 ^b^	0.38 ^b^
149	B	0.32 ^b^	0.39 ^b^	0.32 ^b^	0.33 ^b^
163	B	0.32 ^b^	1.00 ^a^	0.41 ^b^	0.46 ^b^
218	B	0.25 ^b^	1.00 ^a^	0.47 ^b^	0.48 ^b^

^a^ Zero enzyme activity; ^b^ very strong enzyme activity.

## Data Availability

Data are contained within the article.

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
