# Peer review of "Phenotypic and Molecular Characterization of Candida albicans Isolates from Mexican Women with Vulvovaginitis"

_jof, 2025, doi:10.3390/jof11050354_

Round 1
Reviewer 1 Report
Title and abstract
I suggest including a statement or a word that highlights the study was conducted using samples from the Mexican population.
Introduction
L74 – What about the ability to undergo filamentation and form biofilms? Aren’t these important virulence factors in VVC? Please consider discussing them.
L91 – Please elaborate on the potential clinical or epidemiological implications of identifying a strong association between ABC genotypes and enzymatic activity.
Materials and Methods
Inclusion criteria – How many clinical manifestations were required for the diagnosis of VVC? Was the presence of at least one symptom sufficient, or were multiple symptoms needed? Please clarify.
L130 – What were the specific criteria for determining germ tube formation? How many cells were observed per sample? Please provide these methodological details.
Discussion
Please consider including literature that discusses the financial burden of treating VVC to highlight the economic impact of these infections.
It is important to discuss the role of biofilm formation and morphogenesis (filamentation) in the pathogenesis of VVC
L372 – What is the proposed hypothesis for the high enzymatic activity observed?
L378 – What is the role of esterase in fungal pathogenesis? Please propose a hypothesis for the elevated esterase activity observed in your samples.
L387 – What is the role of protease in fungal pathogenesis? Please propose a hypothesis for the high protease activity seen. Also, consider discussing the potential relationship between genotypes A, B, C, and E and the expression of SAP1–SAP10 genes.
Introduction
L61 –Please replace “flora” with “microbioma”
L84 – Please italicize C. albicans
Materials and Methods
Please include a reference for the germ tube test methodology
Results
L247, L340, L342, L343 – Please update the nomenclature of Candida glabrata to reflect its current name
Table 1 – Since the study aims to associate virulence factors with genotype, I suggest grouping all isolates from genotype A together, and doing the same for genotypes B C, to facilitate comparative analysis
Author Response
Title and abstract
I suggest including a statement or a word that highlights the study was conducted using samples from the Mexican population.
Answer: We greatly appreciate this comment. We've included the word "Mexican" in the title and the objective (See lines 38,100).
Introduction
L74 – What about the ability to undergo filamentation and form biofilms? Aren’t these important virulence factors in VVC? Please consider discussing them.
Answer: Thank you for this important observation. Of course, filament and biofilm formation are important factors in the development of VVC, so we've included this information in the Introduction (See lines 75-80).
L91 – Please elaborate on the potential clinical or epidemiological implications of identifying a strong association between ABC genotypes and enzymatic activity.
Answer: A statement was included to briefly outline the clinical and epidemiological implications (See lines 96-98).
Materials and Methods
Inclusion criteria – How many clinical manifestations were required for the diagnosis of VVC? Was the presence of at least one symptom sufficient, or were multiple symptoms needed? Please clarify.
Answer: For the definitive diagnosis of VVC, a positive culture for Candida, with one sign or symptom (white vaginal discharge with no odor, pruritus, burning, dysuria, edema, erythema and/or dyspareunia) was considered. This criterion was included in the manuscript (See lines 115-118).
L130 – What were the specific criteria for determining germ tube formation? How many cells were observed per sample? Please provide these methodological details.
Answer: The criterion for germ tube positivity was the observation of at least five germ tubes in the entire wet mount. Negative results were confirmed by examining at least 10 fields to exclude the presence of germ tubes. Methodological details were included in Material and methods (See lines 139-145).
Discussion
Please consider including literature that discusses the financial burden of treating VVC to highlight the economic impact of these infections.
Answer: Two more references were included that address the topic (see lines 330-333).
It is important to discuss the role of biofilm formation and morphogenesis (filamentation) in the pathogenesis of VVC
Answer: The role of biofilm formation and morphogenesis in the pathogenesis of VVC was discussed (See lines 341-347).
L372 – What is the proposed hypothesis for the high enzymatic activity observed?
Answer: A hypothesis was included for the high hemolysin activity observed in the isolates studied (See lines 392-395).
L378 – What is the role of esterase in fungal pathogenesis? Please propose a hypothesis for the elevated esterase activity observed in your samples.
Answer: Esterases hydrolyze lipid compounds of host cell membranes, thereby causing cell dys-function and disruption, and they likely facilitate Candida adherence and tissue penetration. This information was included in the manuscript, in addition to a hypothesis about the high esterase activity in the isolates analyzed ((See lines 402-407).
L387 – What is the role of protease in fungal pathogenesis? Please propose a hypothesis for the high protease activity seen. Also, consider discussing the potential relationship between genotypes A, B, C, and E and the expression of SAP1–SAP10 genes.
Answer: Proteases participate in the initial steps of colonization by damaging host tissues and di-gestion of the host proteins, such as collagen, keratin, and mucin, to generate nutrients for Candida, which helps the fungus invade the vaginal epithelium more deeply. This information was included in the manuscript (See lines 415-424). However, the discussion of the potential relationship between SAP genes and genotypes was not included because we believe our results do not support a hypothesis in this regard, as genotypes C and E were not found, and genotypes B were rare.
Introduction
L61 –Please replace “flora” with “microbioma”
Answer: The term flora was changed to microbiome (See line 62).
L84 – Please italicize C. albicans
Answer: C. albicans was written in italics (See line 88).
Materials and Methods
Please include a reference for the germ tube test methodology
Answer: Reference was included (See line 145).
Results
L247, L340, L342, L343 – Please update the nomenclature of Candida glabrata to reflect its current name.
Answer: The nomenclature has been updated (See lines 174, 259, 360, 362, 363). The nomenclature for C. krusei, C. guilliermondii, and C. lusitaniae has also been updated (See lines 175, 177).
Table 1 – Since the study aims to associate virulence factors with genotype, I suggest grouping all isolates from genotype A together, and doing the same for genotypes B C, to facilitate comparative analysis
Answer: In the same table, isolates of genotype A and genotype B were grouped together to facilitate comparison (See lines 301-302).
Reviewer 2 Report
The research report by Hugo Díaz-Huerta provides some insights into the molecular and phenotypic features of Candida albicans isolates from women from Mexico.
I must acknowledge that in this paper, the authors do not report the results of the advanced molecular assay of C. albicans, such as genomic or transcriptomic sequencing, which is the limitation of the study. However, they report the results of a reliable assay, which is ABC genotyping of C. albicans, and most importantly, the authors are transparent with their statistical analysis.
I think that the paper deserves to be published, as it provides significant results for Mexican healthcare, and this data and the report can be valuable for the comparative epidemiological analyses of vulvovaginitis around the globe. However, the authors need to address some issues before a decision on the paper.
1. Figure 1. Personally, I do not like pie charts in most cases, but I think that reformatting this figure as a pie chart will be a better option than bar plots because of the title of the Y-axis. The title of the Y-axis is called "Number of isolates", while the first bar represents cases with "No isolation" which is not logical. Please, consider either rewriting the title of the Y-axis or shaping the graph as a pie chart.
2. L. 220-224. Were there samples where yeasts and bacteria were isolated at the same type?
3. L. 210-211. The authors state that there was non-normally distributed data, to which they applied the median and interquartile ranges as descriptive statistics. However, I have not found the report of non-normally distributed data in the results section using these statistics. Moreover, later in this paragraph (L. 213-216), the authors state that they used parametric tests for the hypothesis testing (e.g. ANOVA with post-hocs), which raises the question of whether there were any non-normally distributed data in the reported study. I recommend the authors clarify this, as this text in its current state could be misleading.
4. L. 404-410. Please consider expanding this section with a discussion on how the application of high-throughput molecular assessment methods could enhance the understanding of molecular features of studied isolates.
5. L. 412-415. Please highlight that these results can be applied only to the studied population of Mexican women with vulvovaginitis.
Author Response
The research report by Hugo Díaz-Huerta provides some insights into the molecular and phenotypic features of Candida albicans isolates from women from Mexico.
I must acknowledge that in this paper, the authors do not report the results of the advanced molecular assay of C. albicans, such as genomic or transcriptomic sequencing, which is the limitation of the study. However, they report the results of a reliable assay, which is ABC genotyping of C. albicans, and most importantly, the authors are transparent with their statistical analysis.
I think that the paper deserves to be published, as it provides significant results for Mexican healthcare, and this data and the report can be valuable for the comparative epidemiological analyses of vulvovaginitis around the globe. However, the authors need to address some issues before a decision on the paper.
- Figure 1. Personally, I do not like pie charts in most cases, but I think that reformatting this figure as a pie chart will be a better option than bar plots because of the title of the Y-axis. The title of the Y-axis is called "Number of isolates", while the first bar represents cases with "No isolation" which is not logical. Please, consider either rewriting the title of the Y-axis or shaping the graph as a pie chart.
Answer: We appreciate the comment. The graph has been modified (See lines 238-239).
- L. 220-224. Were there samples where yeasts and bacteria were isolated at the same type?
Answer: Please excuse us, we're not clear on the question. As for the bacteria, we can't determine if they were of the same type because we didn't identify them. As for the yeasts, as mentioned in the Results section, we only identified two species: C. albicans and N. glabratus. The coexistence of yeast and bacteria in the same sample was not observed. Either yeast or bacteria were isolated, but not both types, in the same sample. We try to clarify this in the text (See lines 235-236).
- L. 210-211. The authors state that there was non-normally distributed data, to which they applied the median and interquartile ranges as descriptive statistics. However, I have not found the report of non-normally distributed data in the results section using these statistics. Moreover, later in this paragraph (L. 213-216), the authors state that they used parametric tests for the hypothesis testing (e.g. ANOVA with post-hocs), which raises the question of whether there were any non-normally distributed data in the reported study. I recommend the authors clarify this, as this text in its current state could be misleading.
Answer: Thank you for the opportunity to clarify this. In fact, you are correct that no information on non-normally distributed data was found. In the first draft we were writing, we included clinical information from the study population that had this type of data; however, we considered refocusing the paper, so this data was no longer necessary. Because of this, we have corrected the statistics section in the latest versión (See lines 222-223).
- L. 404-410. Please consider expanding this section with a discussion on how the application of high-throughput molecular assessment methods could enhance the understanding of molecular features of studied isolates.
Answer: Limitations section was expanded in response to your observation (See lines 447-450).
- L. 412-415. Please highlight that these results can be applied only to the studied population of Mexican women with vulvovaginitis.
Answer: We fully agree with the observation, so we rewrite the conclusions (See lines 452-455).
Round 2
Reviewer 1 Report
NA
NA